# Textile-Based Sound Sensors (TSS): New Opportunities for Sound Monitoring in Smart Buildings

**Andrea Giglio** [1,*], **Karsten Neuwerk** [2], **Michael Haupt** [2], **Giovanni Maria Conti** [1] and **Ingrid Paoletti** [1]

1   Department of Architecture, Built Environment and Construction Engineering, Politecnico di Milano, 20133 Milan, Italy; giovanni.conti@polimi.it (G.M.C.); ingrid.paoletti@polimi.it (I.P.)
2   DITF Denkendorf, German Institutes of Textile and Fibre Research Denkendorf, 73770 Denkendorf, Germany; karsten.neuwerk@ditf.de (K.N.); michael.haupt@ditf.de (M.H.)
*   Correspondence: andrea.giglio@polimi.it

**Abstract:** Persistent poor acoustic conditions can imbalance humans' psychophysical capabilities. A good acoustic project starts with either correct measurements of the existing acoustic parameters or with the correct hypothesis of new sound conditions. International standards define invasive measurement conditions and procedures that can disturb user activities. For this reason, alternative methodologies have been developed by mounting real-time sound-monitoring devices. Most of the research on these aims to decrease their dimensions in order to be placed in the tight service spaces of modern architecture and to reduce their aesthetic impact on interiors design. In this perspective, this article explores the features and potentialities of textile-based sound sensors (TSS) as they can not only fulfill these needs but can also be used as architectural ornaments by partially wrapping interiors. The ubiquitous of e-textiles for wearable applications has led to increasing the performance of TSS. Therefore, a comparison of the sensitivity values, signal-to-noise ratio and noise floor of sound TSS with sound sensors is presented, which is still missing in the literature. The paper demonstrates how these can be exploited for sound monitoring and can provide valid opportunities for new smart acoustic textiles.

**Keywords:** e-textiles; textile-based sound sensors; sound monitoring; smart building

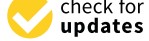

## 1. Introduction

Sound conditions can have physiological and psychological effects on people in either a positive or negative way. Studies have demonstrated effects in educational spaces [1], working spaces [2], restaurants [3], canteens [4] and outdoor spaces [5]. Negative conditions can cause permanent hearing damage, increased stress [6], reduced efficiency at work [7], disturbance of sleep patterns [8] and interference in communications [9] as well as cardiovascular illnesses [10].

Therefore, it is increasingly compelling to embed sound condition considerations starting in the first stages of the design process. International standards indeed provide the requirements for running considerations at the design stage of new programs [11], and for existing functions, international standards give the terms, definitions, measurement conditions, procedures and evaluation methodologies [12–14]. Nonetheless, most of these conditions are invasive and concern the use of annoying sources that can affect normal user activities. For this reason, non-invasive sound-monitoring devices have been developed to provide real-time sound-data gathering.

These systems are framed in the consolidated trends of monitoring systems that help to maximise the energy savings, comfort and safety for the occupants [15]. The most commonly used sensing systems detect real-time information on the temperature and air quality and enable building environmental control systems, such as heating, ventilation, and air conditioning systems if the gathered data does not fulfill the fixed benchmarks.

Indoor environmental quality influences occupant productivity and health; therefore, its management is crucial, and sound has a prominent role. Sound-monitoring devices find applications in urban contexts [16] with low-cost [17] wireless sensor networks [18] embedding spatial statistical analysis [19], through considering indexes that embed subjective measures [20], in detecting noise levels while being powered by energy harvesting [21] and through the development of sensor nodes for accurate indoor sound-level measurements [22].

This article intends to compare the performances of sound sensors of the aforementioned cases with textile-based sound sensors (TSS). The combination of electronics and textiles is increasingly achieving important results in the trend for common computing. The rapid advancements of science and technologies keep revolutionizing traditional textiles and achieving new applications; smart textiles have more functions, including tactile sensing [23], displays [24], communicating [25], self-charging [26] and regulating body temperature [27] and humidity [28].

In the last decades, important advances have been achieved in sound sensors as well. They can measure correlation between the subjective assessment of perceived sound quality and the cardiac activity of the listeners [20], they can be embedded in gloves [29] and used for sound-direction detection, acoustic communications and heart-sound auscultation [30]. Their applications are mostly in wearable fabrics. Greinke used piezo electric films for sound measurement by sticking the film into fabric and sewing a conductive path with conductive yarn [31].

Nakad et al. designed and implemented a large-scale e-textile that functions as an acoustic beamforming array [32]. The prototype aims to find the location of a passing vehicle based upon the vehicle's acoustic emissions through a system that combines multiple lines-of-bearing to the vehicle's traffic. These lines-of-bearing are computed using an acoustic beamforming algorithm under the assumption that the vehicle is in the far field and lies in the same plane as the e-textile. They concluded that, at the physical scale, robust connectors are necessary for attaching electronic components, in order to insulate conductive elements in the fabric. At the system scale, the communication is energy-efficient and fault-tolerant and can serve a wide range of e-textile applications.

Comparisons between the performances of sound sensors have already been presented in the literature; however, a deep comparison of sound sensors for monitoring systems and TSS is still missing. This article intends to fill this gap.

## 2. Research Methodology

The article intends to compare the microphones embedded in sound-monitoring systems with TSSs. Only case studies published in scientific journals or accredited conference are reported.

An imperfect selection of materials, transducer modes and processing circuity can lead poor acoustic functionalities by decreasing the values of the signal-to-noise ratio (SNR) [33–35]. The SNR depends on the sensitivity and noise measured at a given acoustic pressure. The sensitivity is the converted sensing signal arising from the mechanical vibration with respect to the applied acoustic pressure. Therefore, the higher the value is, the better the performance. Furthermore, when the energy conversion takes place, an electrical noise—in form of a voltage/current—is also introduced followed by the thermal-mechanical or Johnson noise [33,34,36]. As a consequence, the signal-to-noise ratio (SNR), sensitivity and noise floor (plotted in Table 1 with the related unit) are the parameters used for the comparison.

**Table 1.** The main parameters and relative units for acoustic sensor performance.

| Parameters | Unit |
| --- | --- |
| Sensitivity at 1 kHz | mV/Pa, or dB |
| Signal-to-noise ratio (SNR) | dB |
| Noise floor | dB |
| Sensitivity Resonance | Hz |

## 3. Methods to Embed Sound Sensors in Textile

One of the most critical aspects for electronic textiles is the power supply. All electronic components require an energy supply in order to be employed as a stand-alone component. Avoiding conventional batteries important since it makes the garment similar to an ordinary one. Batteries are very bulky to add into the fabric, and they can limit the characteristics of the textile itself. Alternatives are given by flexible solar cells (Silicon Solar Inc, Bainbridge, New York, USA.) and micro fuel cells (Enfucell, Vantaa, Finland) since they can be used as a power supply for electronic devices. Other options are given by materials that are able to transform changing pressures (such as body motions or sound pressures in electric power [35]) using piezo electric materials [36]. The piezoelectricity is the generation of electrical polarization in a material under applied mechanical stresses. When a deformation occurs, it generates an electric charge. Vice versa, under applied charge, the material would deform in response [37].

Piezoelectric material-based sensors offer two main advantages: (1) there is no specific requirement for the input power and (2) a broad dynamic range. For these reasons, most of the cases considered for the purpose of this article are piezoelectric-based acoustic sensors (from here on PAS) and micro-electrotechnical systems (from here on MEMS).

Two methods are mainly used to embed sound sensors in textiles: 1. by attaching external sensors to the textile structures and 2. by embroidering piezo electric threads to create the fabric circuits. The two methodologies are described in the next paragraphs.

### 3.1. Piezoelectric Acoustic Sensors (PAS) Films

According to the plane in which the sensing phase is placed, the energy conversions in piezoelectricity takes place in two transducing modes (Figure 1). In the case where it occurs in plane 31, we consider the D31 mode. If it occurs in direction 3, it is the D33 mode.

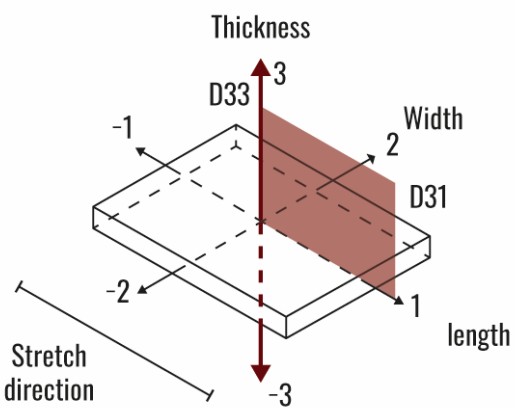

**Figure 1.** Definition of the planes where the sensing phases can be placed.

Upon an application of the acoustic pressure, the induced voltage (V) is defined as [34],

$$V = E \times s \tag{1}$$

where E is the induced electric field and *s* is the electrode spacing. The electrode spacing (s) can be "t" or "d" (as shown in Figure 2), depending on the transducer mode. In D31

mode, the "3-1" refers to the induced polarization in direction 3 with respect to the per unit stress applied in direction 1 (Figure 2a) [35]. Thus, the thickness (t) limits the sensing signal. The improvement of the SNR in order to fit it with new technology demands has been achieved by incorporating different piezoelectric materials with different thicknesses, such as aluminium nitride (AlN) and D31 mode [36], sputtered zinc oxide (ZnO) and D31 mode [38] as well as lead zirconate titanate (PZT) and D31 mode [39].

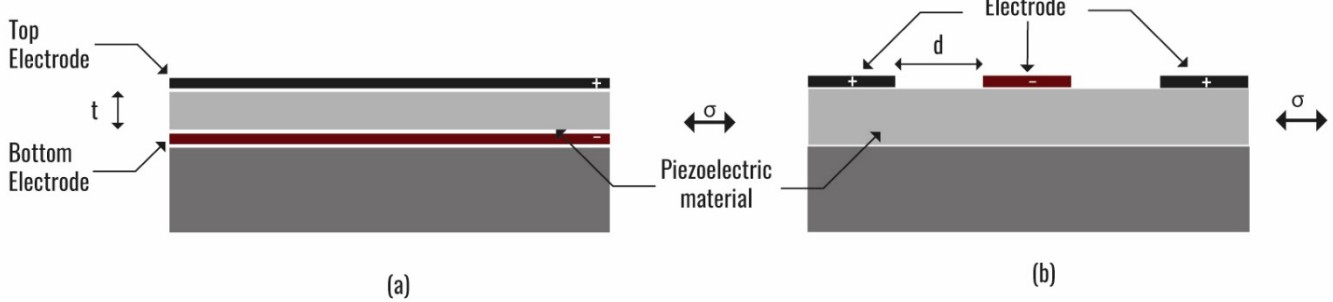

**Figure 2.** Piezoelectric acoustic sensor (PAS) with (**a**) D31 mode where t is the film thickness and (**b**) D33 mode where d is the electrode spacing [40].

The highest thickness values can increase the induced voltage; however, at the same time, it makes the sensor bulky and generates higher noise followed by mass and gravitational acceleration. For instance, Wang et al. reported a 60 dB SNR and a 34 dB noise floor by utilizing 0.7 μm thick PZT [39]; whereas, the reported sensitivity was only 0.49 mV/Pa, which is still a challenge for further signal processing. Therefore, an increase in film thickness is not a favourable solution to enhance the SNR.

For this reason, the D33 mode, where the stress and strain take place in the "3-3" directions (Figure 2b), has been developed. In this mode, the design focuses on the electrode spacing because this mode is not related to the film thickness [40]. Shen et al. incorporated inter-digitated electrodes (D33 mode) to enhance the sensitivity of the piezo electric sensors (PAS) [40]. However, they did not obtain substantial improvement on sensitivity using 250 μm electrode spacing as the reported sensitivity was 0.126 mV/Pa which is far behind than what Wang et al. achieved using D31 mode [39]. The main drawback of the Shen et al.'s PAS is the material selection.

The zirconate titanate (PZT) is good for energy scavenging due to its high piezoelectric constant when compared to other piezoelectric materials [41]. However, the incorporation of zirconate titanate (PZT) requires a special attention during the fabrication, i.e., for the poling process [42]. This is an additional drawback of zirconate titanate (PZT) because it shows the de-poling effect at higher electrode spacing starting from 16 μm which is 4 μm less than the nominal spacing of AlN, i.e., 20 μm [38,43,44].

As a result, the electric field as well as the sensing signal is lower at higher electrode spacing [43]. When compared to the PZT and D33, the combination AlN and D33 can be the better option for acoustic sensors. This combination does not show the de-poling effect since it depends on the lattice orientation [42]. Table 2 categorizes the previous studies by thickness, material, typology, sensitivity, SNR, noise floor and sensitivity peak. Others cases studies are plotted in Table 2 such as [44–47] for piezoelectric acoustic sensors in D31 mode. [48,49] refer to piezoelectric acoustic sensors in D33 mode and are plotted in Table 2.

**Table 2.** References on D31 and D33 piezo acoustic sensors and the relative achieved values. NR: Not reported.

| Reference | PAS Thickness [μm] | PAS Material | Sensitivity mV/Pa at 1 kHz | SNR dB at 1 kHz | Noise Floor | Sensitivity Peak kHz |
|---|---|---|---|---|---|---|
| | | | **D31 mode** | | | |
| [36] | 3 | AlN (Aluminium nitride) | 1.82 | 57 | 37 in dB(A) (A-weighted decibels expresses the relative loudness of sounds in air as perceived by the human ear) | NR |
| [40] | 127 | PZT (lead zirconate-titanate) | 0.12 | NR | NR | 13.71 |
| [44] | 3.5 | ZnO (Zinc oxide) | 0.92 | 37 | 57 Hz (resonance frequency) | 18 |
| [45] | 2.14 | AlN (Aluminium nitride) | 0.039 | 54 | 40 in dB SPL | 20 |
| [46] | 0.267 | PZT (lead zirconate-titanate) | 0.00166 | 58.3 | 35.7 in dB SPL (Sound pressure level measured in decibel) | 59 |
| [47] | 0.2 | AlN (Aluminium nitride) | 0.68 | NR | NR | 11.2 |
| | | | **D33 mode** | | | |
| [48] | 26 | PP (polypropylene) | 2 | 57 | 37 in dB SPL | NR |
| [49] | 0.5 | AlN (Aluminium nitride) | 4.49 | 67 | 27.3 in dB SPL | 10.18 |

*3.2. Piezoelectric Acoustic Sensors (PAS) Yarns*

The studies on piezoelectric yarns aim at overcoming their inability to control changing properties over a wide range of frequencies [50]. This has led to the breach of new applications fields, such as energy harvesting [51] and conformal acoustics [52].

The development of fabrication methods that facilitate multi-functional and multi-material yarns enable several attractive properties for new applications. The preform-based thermal drawing process offers a scalable means of producing kilometre-long fibre devices with sub millimetric cross-sectional dimensions [53]. These long and flexible fibres can easily be assembled into fabrics [54]. Furthermore, the integration of electrodes into the fibre enables the straightforward electrical connection of the device to an external electrical circuit [55].

The latest advancements in this field have led to the development of piezoelectric fibres (based on polyvinylidene fluoride (PVDF) and its copolymer, polyvinylidene fluoride-trifluoroethylene, (PVDF-TrFE)), which are capable of emitting and detecting sound waves over a broad range of frequencies [50]. While the small cross-sectional area of these fibres enables both miniaturization and flexibility, it seemingly involves an equally small active area that potentially limits the fibre performance [52]. The research of Yan et al. overcame the traditional use of acoustic fabrics by introducing a fabric that operates as a sensitive audible microphone while retaining the traditional qualities of fabrics, such as machine washability and draping [30].

Key to the fibre sensitivity is an elastomeric cladding that concentrates the mechanical stress in a piezocomposite layer with a high piezoelectric charge coefficient, of approximately 46 picocoulombs per newton, due to the thermal drawing process [56]. With the fibre subsuming less than 0.1% of the fabric by volume, a single fibre draw enables tens of square metres of fabric microphone. The measured sensitivity of the fibre-on-membrane is 19.6 mV (at 94 dB and 1 kHz, making it comparable to that of off-the-shelf condenser and dynamic microphones [30].

**4. Sound Sensors Typologies in Monitoring System**

The advancements in low-power computing, microphone technology and networking have allowed to move from very expensive static acoustic sensors to low-cost easy-to-use ones. Dedicated stations have been upgraded with real-time data transmission capabilities, but the most important advancements have been achieved in elaborating more flexible sensor node that can perform advanced digital signal processing.

Mydlarz et al. identified three general categories by relating sensor functionality and cost [16]. For the purposes of this article, we considered their sound performance related to the dimensions. When the transfer factor is not specified, the following formula is used to calculate it from sensitivity values in dB re. V/Pa (decibel relative to 1 volt per 1 pascal:

$$\text{transfer factor} \ = 10^{\left(\frac{\text{sensitivity}}{20}\right) \times 1000} \tag{2}$$

If only the transfer factor is presented, the sensitivity is calculated with the following equation:

$$\text{sensitivity} = 20 \times \log_{10}\left(\frac{\text{transfer factor}}{1000}\right) \tag{3}$$

Table 3 categorizes the case studies according with the typology, application field (Indoor or outdoor), sensitivity, transfer factor, signal-to-noise ratio and thickness. It shows that for sound-monitoring system in free field several typologies of microphone have been tested: from Microelectromechanical system (MEMS) to condenser microphones. In these cases, the sensitivity range from −38 to −42.04 dB re. 1 V/Pa with a thickness between 1700 to 17,600 μm. Since the monitoring of noise pollution is more topical in socio-political debate about the comfort of our cities, there are more explorations on open air applications.

**Table 3.** Comparison of sound sensors typologies, thickness, and relative acoustic performances in monitoring systems.

| Reference Source | Sound Sensors Typology | Application | Sensitivity at 1 kHz, dB re. 1 V/Pa | Transfer Factor mV/Pa at 1 kHz | SNR | Thickness [μm] |
|---|---|---|---|---|---|---|
| [16] | MEMS | Outdoor | −38 | 12.59 | 63 dBA | 11,000 |
| [17] | Monacor MCE-400 | Outdoor | −42.04 | 7.9 | 58 dB | 6700 |
| [18] | Tmote-Invent | Outdoor | −35 | 17.78 | NR | 86,000 |
| [19] | Condenser microphone $\frac{1}{2}''$ C-130 Cesva | Outdoor | −35.14 | 17.5 | NR | 17,600 |
| [20] | ADMP401 MEMS | Outdoor | −42 | 7.94 | 62 dBA | 10,000 |
| [21] | Knowles SPU0410LR5H-QB analog MEMS | Indoor | −41 | 8.91 | 94 dBA | 1700 |

Devices which embed all the hardware has adopted with a sensitivity higher than the previous examples at −41 V/Pa at 1 kHz [13]. The high performance represents an opportunity for furthermore applications, but the high volume represents a limitation respect with the necessity to save the net area of use of interiors from services spaces, such as false ceiling and false walls.

## 5. Comparison of Textile-Based Sound Sensors and Other Microphones

The sound sensors in current monitoring systems range between 1700 and 86,000 μm. The sensitivity coefficients cover a broad range between 7.9 to 17.78 mV/Pa re. at 1 kHz. It can be observed that the typologies of microphone in this case are mainly for environmental measurements except for [13]. In this latter, the low thickness is combined with middle low sensitivity respect to the other (Figure 3). These studies are identified in Figure 3 in gradient of green.

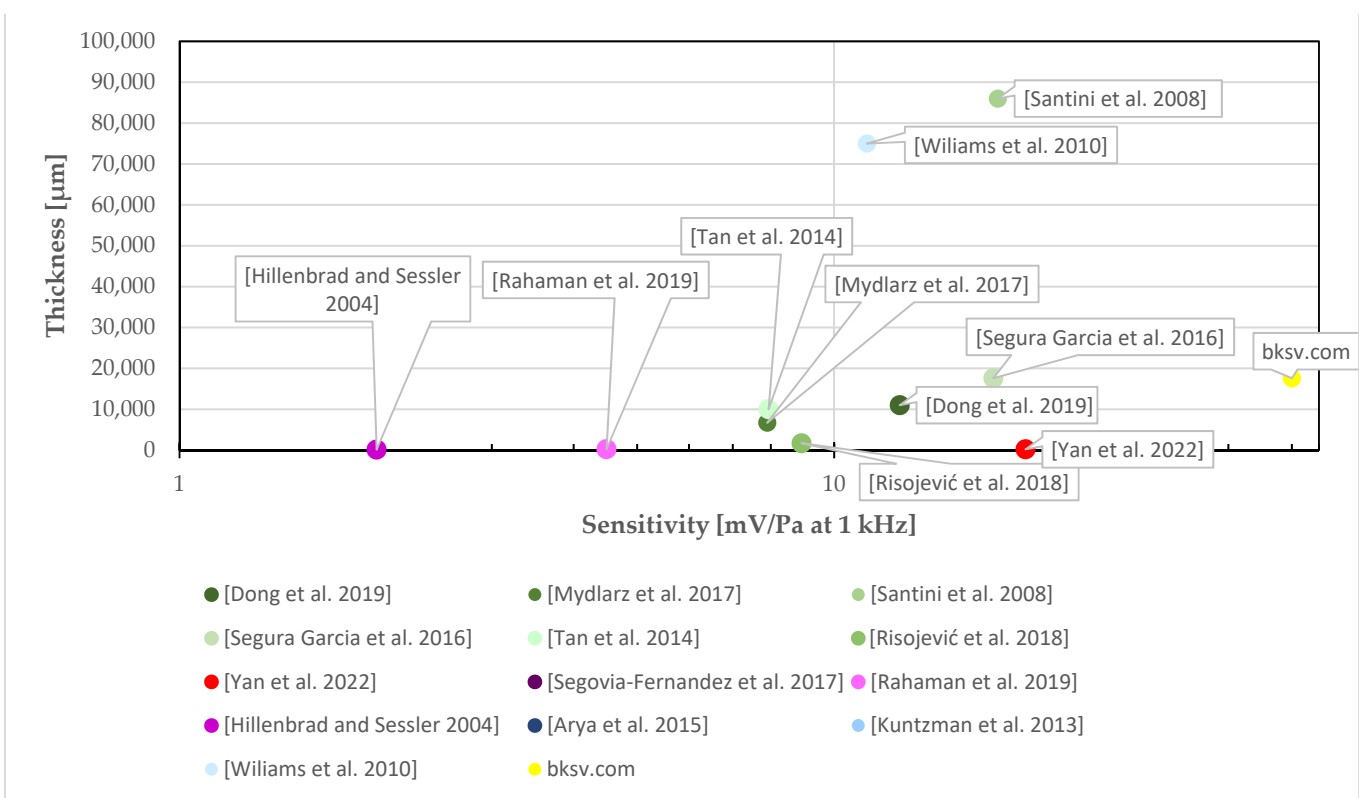

**Figure 3.** Relationship between the thickness and sensitivity of the literature. The sensors of current monitoring systems are in the gradient of green ([Dong et al. 2019] [15], [Mydlarz et al. 2017] [16], [Santini et al. 2008] [18], [Segura Garcia et al. 2016] [19], [Tan et al. 2014] [20], [Risojević et al. 2018] [21]). The PAS yarn is red ([Yan et al. 2022] [30]). The PAS film in D33 mode is the gradient of violet ([Segovia-Fernandez et al. 2017] [47], [Hillenbrad and Sessler 2004] [48], [Rahaman et al. 2019] [49]. The MEMs are in gradient of blue [57], [Arya et al. 2015] [58], [Kuntzman et al. 2013] [59], [Wiliams et al. 2010] [60]) and reference microphone is yellow ([bksv.com] [61]).

Apart from the reference microphones, which have the higher value of sensitivity (in yellow in Figure 3) [61], the studied microphones or sound sensors do not exceed the 19.6 mV/Pa in Yan et al. [30]. This value is achieved by a piezo electric acoustic sensors yarn [30] (identified with red color in the Figure 3). The PAS films remain in a low thickness range since the higher value does not exceed the 240 μm with a sensitivity of 4.49 mV/Pa [39]. They are represented in gradient of violet in the Figure 3.

The MEMS cover a broad range of sensitivity from 0.61 [57] to 11.22 mV/Pa [59] at 1 kHz. Their thickness is proportional to their sensitivity. It has to be considered that the thickness measure does not take in consideration the dimensions of all the other hardware that monitoring systems require. Amplifier, pre-amplifier, add more volume and consequently weight and mass and are an issue in mounting phase when service spaces, such as false ceiling or false wall, are tight. The TSS balanced the acoustic performance with the possibility to be adapted in aforementioned spaces.

## 6. Conclusions

The article aims to compare the sound performance of sound sensors applied in sound-monitoring systems with textile-based sound sensors. The necessity to provide a continuously informed systems to monitor the sound conditions in a space has led to assessing several typologies of microphones aiming to reduce the dimensions and increase the performance. These are requirements that can be fulfilled by the TSSs in both the film and yarn configurations. Table 4 presents how the sensitivity of textile-based sensors is comparable with MEMS, and in some cases, they even have higher performance. This is

more so apparent, when considering the volume of all the hardware that MEMs require to complete a monitoring system. Figure 4 demonstrates that they would increase it considerably.

**Table 4.** Comparison of parameters between Pas D31, Pas D33, Acoustics MEMs and microphones.

| References Source | Sensitivity mV/Pa at 1 kHz | SNR dBA at 1 kHz | Noise Floor | Thickness μm |
|---|---|---|---|---|
| Sound sensors in current monitoring system | | | | |
| [15] | 12.59 | 63 | NR | 11,000 |
| [16] | 7.9 | 58 | NR | 6700 |
| [18] | 17.78 | NR | NR | 860,000 |
| [19] | 17.5 | NR | NR | 17,600 |
| [20] | 7.94 | 62 | NR | 10,000 |
| [21] | 8.912 | −41 | NR | 1700 |
| PAS (Piezo acoustic sensors) yarn | | | | |
| [30] | 19.6 | 30 | NR | 200 |
| PAS (Piezo acoustic sensors) film D33 mode | | | | |
| [48] | 2 | 57 | 37 in dB SPL | 26 |
| [49] | 4.49 | 67 | 27.3 in dB SPL | 0.5 |
| MEMS | | | | |
| [57] | 1 | NR | 16 in (kHz) | 2.0 |
| [58] | 0.0966–0.1266 | NR | 85 in (kHz) | 15 |
| [59] | 0.61 | NR | 13 in (kHz) | 29 |
| [60] | 11.22 | NR | 176 in (kHz) | 150,000 |
| [48] | 2 | 57 | 37 | 125 |
| Conventional microphone | | | | |
| [61] | 50 | NR | 15 dBA | 17,600 |

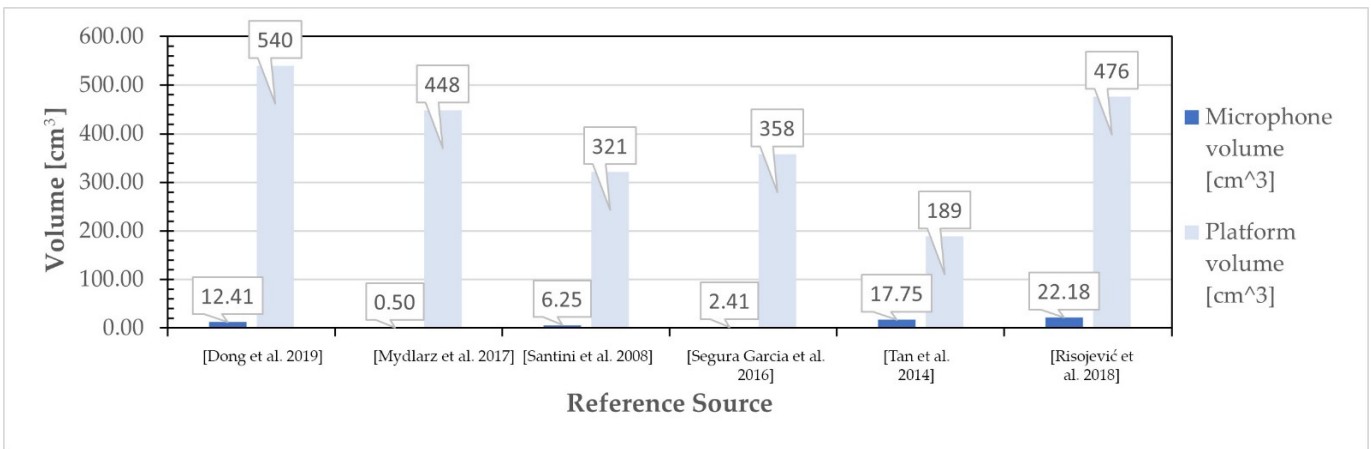

**Figure 4.** The relations between the volume of the MEMS microphones themselves and with the hardware of the monitoring system. ([Dong et al. 2019] [15], [Mydlarz et al. 2017] [16], [Santini et al. 2008], [18], [Segura Garcia et al. 2016] [19], [Tan et al. 2014] [21], [Risojević et al. 2018] [22]).

The low thickness of TSSs can facilitate use in tight service spaces, such as false ceilings and false walls, in order to reduce the aesthetic impact on the space design. Therefore, one of main requests for these systems is either the reduction of their shapes or a high degree of adaptability. Moreover, due to the aesthetic characteristics of the textile itself, in future works it could be interesting to demonstrate how they can be used to combine aesthetic features with sound-monitoring ones.

This feature can enable sound smartness in fabrics with active absorption behaviours to improve sound conditions in real time, thereby, opening new scenarios for the development of acoustic textiles.

**Author Contributions:** Conceptualization, investigation, writing—original draft preparation, A.G.; writing—review and editing, K.N.; supervision, M.H., G.M.C., I.P. All authors have read and agreed to the published version of the manuscript.

**Funding:** This research received no external funding.

**Data Availability Statement:** The data presented in this study are available in the article.

**Conflicts of Interest:** The authors declare no conflict of interest.

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
