# Peer review of "Textile-Based Sound Sensors (TSS): New Opportunities for Sound Monitoring in Smart Buildings"

_textiles, doi:10.3390/textiles2020016_

Round 1
Reviewer 1 Report
Dear Authors,
in your interesting manuscript, the following points should be added/changed to further improve it:
- Generally, if you name a review manuscript as "Article", meaning a report about your own research, this will make reading quite strange for the reviewers (and afterwards for the other readers).
- It is not necessary to number the keywords.
- Fig. 1 is mostly not readable; please make it larger and sharper. Did you receive the permission to republish the figure, or was it originally published under a CC-BY license?
- line 45: before and after what?
- line 46: Please define RT as reverberation time. What are the other parameters in Eq. 2?
- line 50: What is A?
- line 53: And what is r0?
- Eq. 3: What is lg?
- line 56: "This means the effect of sound absorption increases with the distance from the source."- this does not follow from the previous sentence.
- lines 130 and 135: I assume the references should be given in the list of references at the end of the manuscript.
- Fig. 2: Do you have the permissions to republish these images?
- Tables in the appendix shouldn't have the same numbers as tables in the main text.
- line 233: "allow to pass the signal"
- line 244 "Using conventional batteries is the point of is-sue since it makes the garment as similar as ordinary one." - what do you mean here?
- Eqs. 3 and 4: What do all the subscripts mean, and what is the superscript "E"?
- Table 3: Polyvinylidene fluoride, Polyurea
- line 325: CF_2--CH_2 (subscripted)
- Fig. 4 is not sharp and looks stretched.
- Fg. 5 (called Fig. 2) is also not sharp and not mentioned in the text. Generally, please correct the numbering of the figures and check carefully whether they are all correctly mentioned in the text.
- line 456: This sentence doesn't have a verb.
- Table 6: Units are not allowed to be in square brackets.
- Generally, after reading the whole text, I am still wondering what I should have learnt. This may partly be a problem of the wrong expectation due to its label as "Article" as well as of the chaotic numbering and also writing of the last part. Nevertheless, I couldn't tell what this article is about - I don't even know what you mean with the "architectural acoustics applications" mentioned in the title. Please try to find a better "thread" guiding the reader through the text. Please explain in the introduction what you are aiming at, what the text will report on, and what is different to previous reviews about the same topic.
Author Response
Dear reviewer,
thanks for you comments.
you can find the attached point-by-point response.
best.

Reviewer 2 Report
This paper describes the sensors and actuator of e-textiles in acoustic applications, especially for PVDF sensors. In abstract, the author says five scenarios of e-textiles applications for architectural acoustics purposes, there are no explicit five scenarios in the text. The chapter structure of the entire paper should be reviewed so that the reader can better understand it. Chapter 3 describes scenarios for the application of e-textile to architectural acoustics, but additional information needs to be added regarding the dimension, response time, power consumption, stress and other specifications required for the application. I cannot recommend the article to be published in Textiles in the present form. Additionally, please consider the following points.
1. It would be better to open up the line space between items in the table, as they are packed and difficult to read.
2. On page 8, line 301, the table is mis-numbered (7 instead of 3).
3. On page 9, line 317, the space between "Handford" and "and" is missing.
4. After the figure on page 11, the figure numbers are wrong (2 instead of 5). The reference numbers of the figures in the text are accordingly incorrect.
5. On page 12, lines 439 and 446, the character [µ] is missing for "micro-metre".
6. On page 14, line 496, it is better to unify "fibre" rather than "fiber".
7. On page 9, line 339, it might be 40-50 µm instead of milli-metre.
Author Response
Dear Reviewer,
thanks for you comments.
You can find the attached point-by-point response.
Best

Round 2
Reviewer 1 Report
Dear Authors,
after complete rewriting your manuscript, the following points remain open:
- Your manuscript is really not a "journal article", but a review.
- There's a lot of new grammar problems and typos; please have the text corrected by a native speaker or translator.
- Figures: Please mention whether the re-published images were originally published under a CC-BY or a similar license, or whether you received the permittance to re-use them. The second Fig. 1 and Fig. 2 are actually Figs. 2 and 3.
- Table 2: What is "in dB SPL"? Please don't put units into square brackets, this is not allowed according to ISO 80000. And why is there one negative SNR value?
- line 214: Please explain P(VDF-TrFE).
- Fig. 4: What does the small black line on the lower right side mean?
- Eq. 1: The transfer factor cannot have the unit mV/Pa (not in square brackets, please) as long as on the right side no units occur. The exponent is also problematic since it cannot have a unit. And what do you mean with "dBre.1V"?
- Similarly, the sensitivity in Eq. 2 cannot have a unit as long as the right side does not have a unit, and the argument of the log must not have a unit.
- Table 3: If the noise floor is never given, it does not make sense to have a column for it.
- Fig. 5: No units in square brackets, please.
- Table 4: What does Pas mean, should it be PAS? The units should be separated by the physical parameters in the usual ways, either by putting them into round brackets, behind a slash or behind "in".
- line 294: Table 6 is Table 4.
Author Response
Dear Reviewer,
thanks for your comments and feedback.
The journal "Textiles" and this special issue invites to apply "research articles, review articles as well as short communications" as you can read from the "Manuscript submission information" paragraph at this link https://www.mdpi.com/journal/textiles/special_issues/trends_textiles. I don't know how to change the type in the description format of the manuscript. I will contact the editor to change it. Then, since we did not buy the permission to publish the pictures, we have decided to not use the pictures but to quote the relative research in the text.
Please in the attachment you can see these modifications:
- A translator checks the text.
- We delated the pictures since we did not buy the permission to publish them
- Figures numbering are correct
- Table 2: I have deleted the square brackets and correct the negative SNR value
- I have explained P(VDF – TrFE)
- 4 is deleted
- Eq 1.: I have deleted the units in the equation, and I have explained unit “dB re. 1V” in the text.
- 2: I have deleted the units
- Table 3: I have deleted the column of noise floor
- 5: I have deleted the square brackets
- Table 4: I have corrected PAS adding the explanation version as well. And I have corrected the units in the cells
- Line 294: I have corrected in Table 4

Reviewer 2 Report
This paper describes a review of textile-based sound sensors (TSS) for sound monitoring system. The paper provides an overview of previous research on textile sound sensors and compares their sensitivity, responsiveness, and size structure. Please consider the following points before publication.
1. On page 4, line 156, the figure is mis-numbered (2 instead of 1).
2. On page 5, line 175, the figure is mis-numbered (3 instead of 2).
3. On page 7, line 229, it is better to explain the black line (bar) at the right bottom.
4. On page 9, line 284, it might be a typo at the end of the sentence.
5. on page 9, line 288, the numbering of the chapter "Conclusions" is 5 instead of 6.
6. On page 9, line 294, the table number is 4 instead of 6 in the text.
Author Response
Dear Reviewer,
thanks for your comments and feedback.
- I have provided new numbering since some of them have been deleted.
- I have provided new numbering since some of them have been deleted.
- I deleted the picture
- I have corrected the typo
- Now the paragraph “Conclusion” has corrected numbering.
- I have corrected the table number.
Best regards.